# Contribution of apical and basal dendrites to orientation encoding in mouse V1 L2/3 pyramidal neurons

Jiyoung Park[1,2,7]*, Athanasia Papoutsi[3,7], Ryan T. Ash[1,4,5], Miguel A. Marin[4,6], Panayiota Poirazi [3,8]* & Stelios M. Smirnakis[1,8]*

Pyramidal neurons integrate synaptic inputs from basal and apical dendrites to generate stimulus-specific responses. It has been proposed that feed-forward inputs to basal dendrites drive a neuron's stimulus preference, while feedback inputs to apical dendrites sharpen selectivity. However, how a neuron's dendritic domains relate to its functional selectivity has not been demonstrated experimentally. We performed 2-photon dendritic micro-dissection on layer-2/3 pyramidal neurons in mouse primary visual cortex. We found that removing the apical dendritic tuft did not alter orientation-tuning. Furthermore, orientation-tuning curves were remarkably robust to the removal of basal dendrites: ablation of 2 basal dendrites was needed to cause a small shift in orientation preference, without significantly altering tuning width. Computational modeling corroborated our results and put limits on how orientation preferences among basal dendrites differ in order to reproduce the post-ablation data. In conclusion, neuronal orientation-tuning appears remarkably robust to loss of dendritic input.

[1] Brigham and Women's Hospital and Jamaica Plain VA Hospital, Harvard Medical School, Boston, MA, USA. [2] Program in Structural and Computational Biology and Molecular Biophysics, Baylor College of Medicine, Houston, TX, USA. [3] Institute of Molecular Biology and Biotechnology (IMBB), Foundation of Research and Technology Hellas (FORTH), Vassilika Vouton, HeraklionCrete, Greece. [4] Department of Neuroscience, Baylor College of Medicine, Houston, TX, USA. [5] Present address: Department of Psychiatry and Behavioral Sciences, Stanford University, Stanford, USA. [6] Present address: Department of Neurology, University of California, Los Angeles, USA. [7] These authors contributed equally: Jiyoung Park, Athanasia Papoutsi. [8] These authors jointly supervised this work: Panayiota Poirazi, Stelios M. Smirnakis.  *email: jiyoung.p.neurons@gmail.com; poirazi@imbb.forth.gr; smsmirnakis@bwh.harvard.edu

Neocortical pyramidal neurons ramify several basal dendritic arbors laterally and one apical dendritic arbor superficially to receive and integrate synaptic inputs[1]. Each dendritic-tree of a mouse layer 2/3 (L2/3) pyramidal neuron arborizes in a different cortical map sub-region (e.g., retinotopic region) and/or cortical layer, thereby sampling largely non-overlapping axonal inputs coming in from different brain areas[2]. Inputs are functionally heterogeneous across dendrites and even within individual dendritic branches[3,4]. Mouse primary visual cortex (V1) L2/3 pyramidal neurons generate action potentials in response to a narrow range of orientations[5] despite receiving highly heterogeneous input and poorly tuned subthreshold responses[4], making them ideal for studying the relationship between dendritic input and functional selectivity. Indeed, recent evidence suggests that apical tuft dendritic spikes serve to narrow the orientation tuning function, increasing orientation selectivity of area V1 L2/3 pyramidal neurons[6]. L2/3 pyramidal neurons are also a good model system for studying the relative roles of apical and basal dendrites: while basal dendrites primarily receive feedforward input from L4 and nearby L2/3 neurons[7,8], apical dendrites receive cortico-cortical feedback that may refine orientation selectivity[9,10] as well as orientation-tuned thalamo-cortical input from layer 1[4,11–13].

It is difficult to firmly establish a causal relationship between computations that occur in dendritic branches and properties at the soma. Calcium imaging does not always have the temporal resolution to disambiguate signals arising in different dendritic processes or the soma, while in vivo dendritic patch clamping, despite its power at dissecting specific hypotheses, is usually limited to assessing single branch contributions to neuronal response. The existence of functional inputs and activity on dendrites does not causally explain the necessity of these inputs for determining the neuron's final output. Here we employ in vivo two photon microdissection[14–17] to systematically remove individual dendrites from layer 2/3 mouse V1 pyramidal neurons, allowing us to assess the causal relationship between inputs arriving in different dendritic arbors and the computation of orientation selectivity at the soma.

## Results

**Dendrite ablation in vivo.** To assess the causal relationship between inputs arriving in different dendrites and orientation-tuning, we systematically removed individual dendrites from L2/3 mouse V1 pyramidal neurons using in vivo 2-photon micro-dissection[14,15,17]. We visualized both dendritic structure and functional activity of L2/3 pyramidal neurons by stereotactically co-injecting AAV-flex-GCaMP6s and diluted AAV-CaMKII-Cre (40,000 to 120,000×) into L2/3 of mouse V1[11] (Fig. 1a, methods). Individual dendrites were then removed by performing 2-photon laser point scans on the targeted fluorescent dendritic segment, 10–30 μm away from the soma (Fig. 1b, c and Supplementary Fig. 1)[14,15]. Dendritic segments expressing GCaMP6s, located 130–200 μm from the dura, could be typically ablated with one or two 200–400 ms point scans of 140–200 mW at 800 nm. Ablated dendrites distal to the lesion acquired a beads-on-the-string appearance and degraded, disappearing within 1–3 h (Fig. 1b, Supplementary Movie 1,2)[14,15]. Upon ablation, target neurons transiently became brightly fluorescent (Supplementary Fig. 1d), presumably due to the influx of calcium through the instantaneous opening in the membrane, returning to pre-ablation fluorescence levels over the next 5–90 min. Neurons whose fluorescence did not return to their pre-ablation levels within 90 min were more likely to be destroyed, i.e., visually undetectable with debris present, by the next day. Neurons whose apical dendritic arbors were ablated immediately adjacent to the soma were also more likely to

disappear, so we restricted ablation to pyramidal neurons with an extended primary apical trunk (≥20–3 μm from soma to the first apical bifurcation), targeting the point immediately prior to the bifurcation. Target neurons had a soma depth between 150 and 250 μm. Neurons that were still present 24 h post-ablation (46% of cells: 18 out of 39 neurons from 26 mice survived after apical dendrite ablation, see "Methods") exhibited intact morphology in their residual (non-ablated) dendritic segments and normal spontaneous and visual-evoked somatic calcium transients (Fig. 1b–d, Fig. 2a and Supplementary Fig. 1).

We confirmed that targeted dendrites were successfully removed by generating a custom virus co-expressing GCaMP6s with activity-independent stable red fluorophore mRuby (Supplementary Fig. 2). The elimination of the apical tuft following ablation was clearly identified with red fluorescent signal, and complete structural overlap was observed between GCaMP6 and mRuby. Ablation was further confirmed using post-hoc immunostaining with anti-GFP antibodies (Supplementary Fig. 3). Post-hoc immunostaining of unlabeled axons and dendrites adjacent to the ablation site revealed no overt signs of damage (Supplementary Fig. 3e, Supplementary Movie 3) while adjacent GCaMP-labeled processes imaged in vivo near the ablation site were not noticeably affected (Supplementary Fig. 1c), suggesting that ablation effects are spatially restricted within a radius of ~5 μm around the ablation point, in line with previous reports[14,15].

**Loss of apical input does not alter orientation preference.** L2/3 pyramidal neurons receive orientation-tuned inputs scattered pseudo-randomly on their apical dendritic-trees[3,4,11,18], but it is not clear how these contribute to the neuron's orientation preference. We tested the collective contribution of these inputs to orientation selectivity by ablating the apical tuft from L2/3 pyramidal neurons in fentanyl-dexmedetomidine sedated mice. The ablation of apical dendrite removes ~40% of the total excitatory input received by the neuron[19]. Figure 1c (and Supplementary Movie 2) shows the reconstructed morphology of one neuron before and after apical tuft ablation. Figure 2a illustrates single-trial dF/F responses of an example neuron to oriented gratings moving in 12 different directions, before as well as 1, 3, and 5 days after apical tuft ablation. Remarkably, removal of the entire apical tuft did not affect the neuron's preferred orientation (Fig. 2b). The ablated neuron still responded maximally to its pre-ablation preferred orientation, and responses to other orientations were also very similar to pre-ablation (Fig. 2c).

Plotting the absolute-value change in orientation preference before and 5 days after ablation demonstrated that orientation preference did not change significantly following apical tuft ablation (Fig. 2d: blue circles corresponding to 18 ablated neurons all fall close to the diagonal as do black squares corresponding to 17 neighboring control neurons imaged together with ablated neurons). There was no significant difference in orientation preference change between the two groups (Fig. 2d, inset; $p = 0.4$, t-test; $p = 0.95$, Mann–Whitney $U$ test). Tuning-width (width at half maximum) and orientation selectivity index (OSI = [Response$_{pref}$ − Response$_{null}$]/[Response$_{pref}$ + Response$_{null}$]) were also not significantly affected on average by apical tuft ablation (Fig. 2e, $p = 0.81$, t-test; $p = 0.37$, Mann–Whitney $U$ test; Fig. 2f, $p = 0.88$, t-test; $p = 0.47$, Mann–Whitney $U$ test). Bootstrap analysis to assess per-neuron shift in orientation preference and tuning width (see "Methods") also suggests that there is no consistent shift in orientation preference after removal of the apical dendrite beyond that observed in controls (Supplementary Figs. 4–5, Supplementary Table 1). The narrow distributions of bootstrap-estimated orientation preferences after ablation indicates that the tuning of neurons remains robust even following

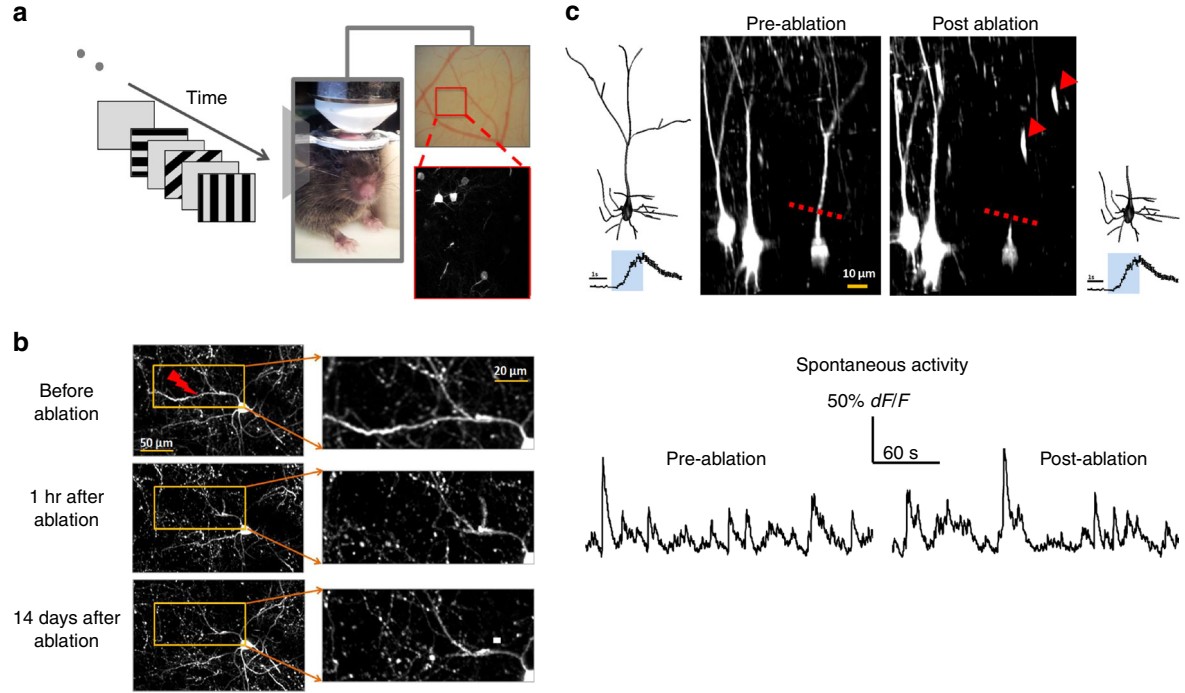

**Fig. 1** In vivo 2-photon laser dissection of apical dendrites in mouse primary visual cortex. **a** Experiment set-up. Left: visual stimulus. Middle: mouse under the objective. Right top: optical image of the surface of V1. Right bottom: two-photon image acquired from the inset showing sparse GCaMP6s-labeled neurons. **b** Dendritic arbor before (top), 1 h (middle) and 14 days (bottom) after targeted laser-dissection of a single dendritic branch of a GFP-expressing neuron (red arrow: ablation point). Right panels zoom in. **c** Center: 3D renderings of the target neuron and its neighboring control neurons before (left) and after (right) apical dendrite ablation. Broken red line: the ablation point. Red arrows: GCaMP filled remnants of the severed apical dendrite. 3D reconstruction of target neuron before (far left) and after (far right) apical dendrite ablation. **d** Spontaneous GCaMP6s activity of the example neuron before and after apical tuft ablation.

apical dendrite ablation (Supplementary Fig. 4). No consistent changes in response amplitude or baseline firing rates were observed across neurons a day or more following ablation (Supplementary Fig. 6, see figure legend for statistics). Pre- and post-ablation tuning curves for all recorded neurons are shown in Supplementary Fig. 7. This demonstrates that, under the conditions tested, inputs to the basal dendrites are sufficient for determining the orientation preference and basic orientation tuning function properties of V1 L2/3 pyramidal neurons[7].

**Orientation tuning is robust to loss of multiple basal dendrites.** We then applied our ablation strategy to the basal dendrites to explore how individual primary basal dendrites contribute to orientation selectivity (Fig. 3). Mouse V1 L2/3 pyramidal neurons typically have 5–8 (median 6) primary basal dendrites[19]. The point scan was targeted to a randomly-chosen primary basal dendrite, 5–10 μm away from the soma, before secondary branching. Neurons still present 24 h post-basal dendrite ablation survived long-term and were studied. Overall, 36% of functionally studied neurons survived basal dendrite ablation (13 out of 33 neurons survived single basal dendrite ablation and 17 out of 50 neurons survived double basal dendrite ablation, from 34 mice). This was not significantly different from the survival rate after apical dendrite ablation (46%; $p = 0.29$, $\chi^2 = 1.1$, chi-square test).

Similar to what we observed with apical dendrite ablation (Fig. 2), cutting one basal dendrite (Fig. 3a, 8–12% of total dendritic input) did not alter the target neuron's preferred orientation, nor its tuning-width or OSI (Fig. 3b, Supplementary Fig. 7b). Neurons in which two basal dendrites were removed (Fig. 3c, 16–24% of total input) also demonstrated remarkable tuning curve stability (Fig. 3d, Supplementary Fig. 7b), but did show a small, yet significant, shift of ~12.5 ± 2.7° on average

(max: 30°) in orientation preference following ablation (Fig. 3e–f, two-basal ablation vs. control: $p = 0.003$, ANOVA with Tukey correction for multiple comparisons; $p = 0.025$, Kruskal–Wallis test with Tukey correction for multiple comparisons. Two-basal-ablation vs. pooled one-basal-ablation + control: $p = 9.5 \times 10^{-12}$, $t$-test; $p = 0.0086$, Mann–Whitney $U$ test). We again performed bootstrap analysis to estimate the reliability of orientation preference measurements neuron by neuron. The distribution of the bootstrap estimates of orientation preference before and after ablation was narrow, suggesting that the tuning properties of ablated neurons were measured reliably (Supplementary Figs. 4–5, Supplementary Table 1). Per-neuron bootstrap analysis also demonstrated that the number of two-basal dendrite ablated neurons that shifted their orientation preference post-ablation was significantly different compared to controls (two-basal: 7/17 neurons significantly shifted, control: 2/35 significantly shifted, $p = 0.0015$, $\chi^2 = 10.1$, chi-square test), while this was not true for apical or one-basal dendrite ablations, which were not significantly different from controls (apical: 3/18 significantly shifted, $p = 0.19$, $\chi^2 = 1.67$; one-basal: 3/13 significantly shifted, $p = 0.08$, $\chi^2 = 3.1$, chi-square test, Supplementary Table 1). Tuning-width and OSI differences did not reach significance (Fig. 3g–h, tuning width: $p = 0.19$; OSI: $p = 0.36$, ANOVA with Tukey multiple-comparisons test; See Supplementary Fig. 8 for histogram distribution of changes in tuning width and OSI; See figure legend for Fig. 3 for Kruskal–Wallis test results). This result further emphasizes that V1 L2/3 neurons' orientation tuning is remarkably robust to loss of dendritic input (Supplementary Fig. 9 and Supplementary Movie 4).

**Modeling tuning robustness of apical dendrite ablation.** Our finding that apical dendrite ablation has no effect on the

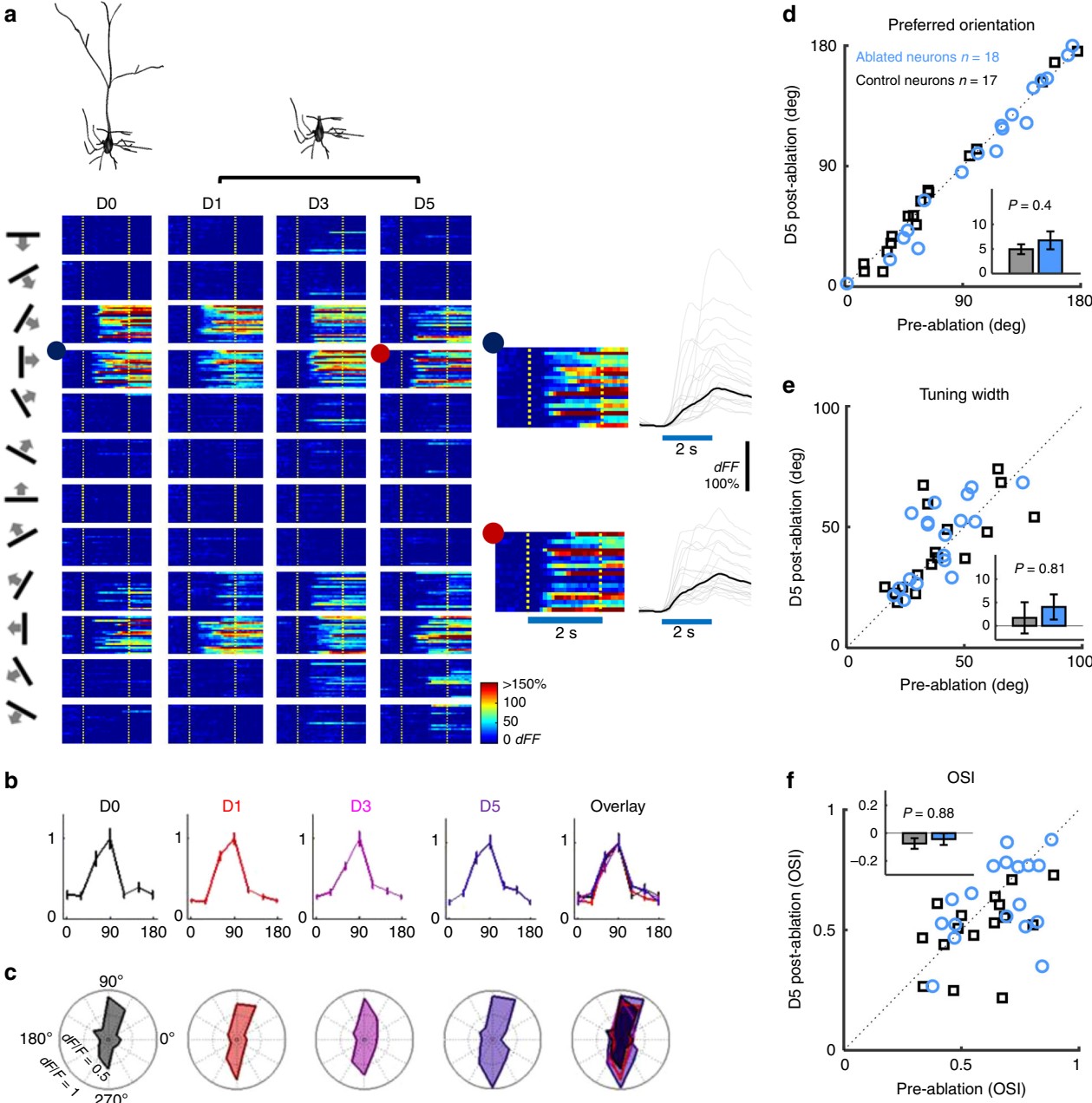

**Fig. 2** Orientation-tuning functions are unchanged following apical dendrite ablation. **a** Top: 3D reconstruction of an example neuron before and after apical dendrite ablation. Bottom: Heat map depictions of the neuron's single trial somatic calcium responses. Each column depicts responses to 12 directions (far left) before (D0) and 1 (D1), 3 (D3), and 5 days (D5) after ablation. Stimulus onset and offset are marked with broken yellow lines. Right insets: Zoomed-in view of the neuron's responses to a rightward moving grating in 30 trials at D0 (blue dot) and D5 (red dot). Blue line: stimulus duration. **b** Peak-normalized orientation-tuning curves of the example neuron at D0 (black), D1 (red), D3 (purple), D5 (blue) and overlay of D0–5. **c** Polar plots depicting un-normalized orientation-tuning curves at each time point. **d–f**, scatter plots of **d** orientation preference, **e** orientation tuning width and **f** orientation selectivity index (OSI) from von-Mises fitted tuning curves of each neuron before and 5 days after ablation. Blue, ablated neurons ($n = 18$). Black, control neurons ($n = 17$). Insets in d–f are bar plots of mean changes in (**d**) orientation preference (absolute-value change), (**e**) tuning width and (**f**) OSI from von-Mises fitted tuning curves of each neuron 5 days after ablation. Error bars are standard error of mean. $P$ values in figure are from t-test.

orientation-tuning curve of L2/3 pyramidal neurons suggests a redundant contribution of the apical compared to the basal tree in orientation encoding. Otherwise, ablation should cause a change in neuronal orientation preference. To explore the possible input structures producing this result, we simulated a morphologically detailed L2/3 pyramidal neuron of mouse V1, with validated active and passive properties[6,20] as well as literature-based synaptic density and single-synapse orientation-tuning properties (Fig. 4a–c and Supplementary Fig. 10, see Supplementary

Data for details of the model neuron)[4,19,21]. Three main parameters were varied: standard deviation ($\sigma_{apical}$) of the distribution of single-synapse orientation preferences ($pref_{syn}$) in the apical dendritic-tree, standard deviation ($\sigma_{basal}$) of the distribution of $pref_{syn}$ in the basal dendritic-tree, and the difference in the mean orientation preference of these two distributions ($\Delta(\mu_{basal}, \mu_{apical}) = |\mu_{basal} - \mu_{apical}|$) (Fig. 4a, Supplementary Fig. 10). We found that to generate tuning curves comparable to experiments (OSI ≥ 0.2, tuning-width ≤ 80°), ($\sigma_{basal}$, $\sigma_{apical}$) cannot be simultaneously

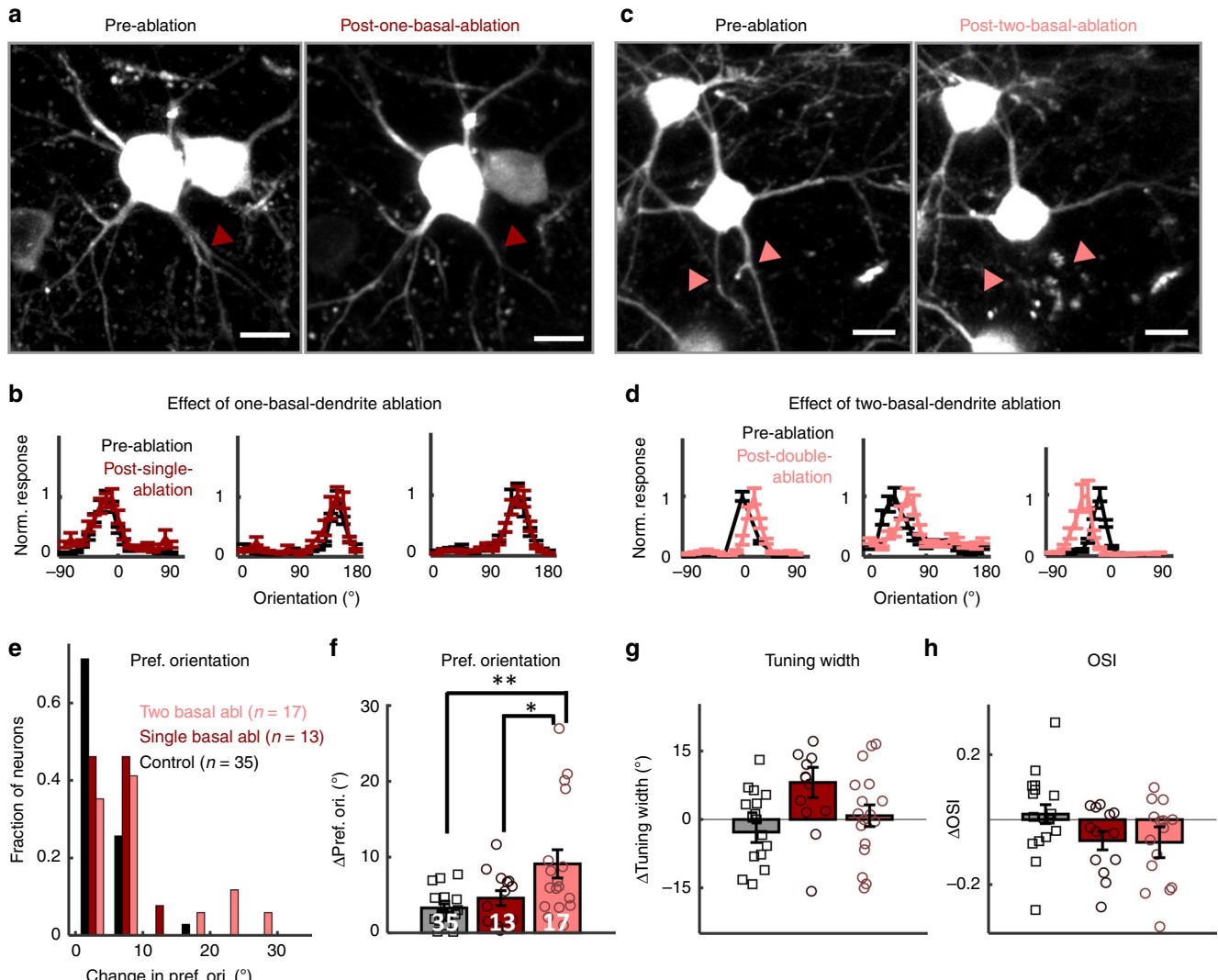

**Fig. 3** Removal of two, but not one, primary basal dendrites shifts orientation preference. **a** Z-stacked top view of an example ablated neuron and a neighboring control neuron before (left) and after (right) a single primary basal dendrite ablation. Ablated dendrite is marked with dark-red arrow. **b** Three examples of peak-normalized orientation-tuning curves before (black) and 5 days after one-basal-dendrite ablation (dark-red). **c**, **d** Figures in the same format with a-b for two-basal-dendrite ablation (marked with pink). Error bars are standard error of mean across trials. **e** Histogram of the absolute-value change in preferred orientation (deg) for control (black, $n = 35$), one-basal-ablation (dark-red, $n = 13$), and two-basal-ablation (pink, $n = 17$) neurons. **f**–**h** Mean changes in (**f**) preferred orientation (absolute-value change, ANOVA: $F(2, 62) = 6.29$, $p = 0.003$, Kruskal–Wallis: $\chi^2(2,62) = 7.03$, $p = 0.029$), **g** tuning width (ANOVA: $F(2, 62) = 1.71$, $p = 0.19$, Kruskal–Wallis: $\chi^2(2, 62) = 1.44$, $p = 0.49$), and **h** orientation selectivity index (OSI, ANOVA: $F(2, 62) = 1.03$, $p = 0.36$, Kruskal–Wallis: $\chi^2(2, 62) = 0.59$, $p = 0.75$) from von-Mises fitted tuning curves of each neuron 5 days after ablation. See Supplementary Fig. 8 for a histogram of the distribution of changes in tuning width and OSI. Error bars are standard error of mean. For (**f**), numbers in white indicate $n$-number of each condition. **$p = 0.003$ (control vs two-basal), *$p = 0.04$ (one-basal vs two-basal), ANOVA with Tukey correction. Scale bars for (**b**) and (**d**) are 10 μm.

larger than 45° (Supplementary Fig. 11a-e). Interestingly, for all $\Delta$ ($\mu_{basal}$, $\mu_{apical}$) ranging from 0° to 90°, the model neuron's orientation preference was consistently biased towards that of the basal tree (Fig. 4d), indicating the relative dominance of the basal dendrites.

To further constrain the input parameter space, we modeled the effects of apical dendrite ablation on preferred orientation, OSI and tuning-width (Fig. 4e, Supplementary Fig. 11f). We set $\sigma_{apical}$ at 30° as approximated from prior measurements of single-synapse orientation preference distribution on apical tuft dendrites (Fig. 4e, Supplementary Fig. 11f)[11]. Figure 4e shows example tuning curves for $\sigma_{apical} = \sigma_{basal} = 30°$ with varying $\Delta$ ($\mu_{basal}$, $\mu_{apical}$). For large $\Delta$($\mu_{basal}$, $\mu_{apical}$) (>40°, Fig. 4e right, Supplementary Fig. 11f upper half of heat-maps), the apical

dendrite input functions as noise, broadening the orientation-tuning curve, and therefore its removal leads to shifts in orientation preference, narrower tuning-width and higher OSI, contrary to experiment. For intermediate $\Delta$($\mu_{basal}$, $\mu_{apical}$) (20°–40°) and $\sigma_{basal} \leq \sigma_{apical}$ (30°), the basal dendrite dominates, and apical dendrite ablation has no effect (Supplementary Fig. 11f), consistent with experiment. For $\sigma_{basal} > \sigma_{apical}$ (30°), the apical dendrite input sharpens orientation-tuning, and therefore ablation of the apical dendrite leads to wider tuning width and lower OSI (Fig. 5f right two panels, Supplementary Fig. 11f). For small $\Delta$($\mu_{basal}$, $\mu_{apical}$) (0–10°, Fig. 4e left panel, Supplementary Fig. 11f lower half of heat-maps), the model neuron's preferred orientation does not change (i.e., shift in orientation preference <10°) following ablation as long as $\sigma_{basal}$ is <60°.

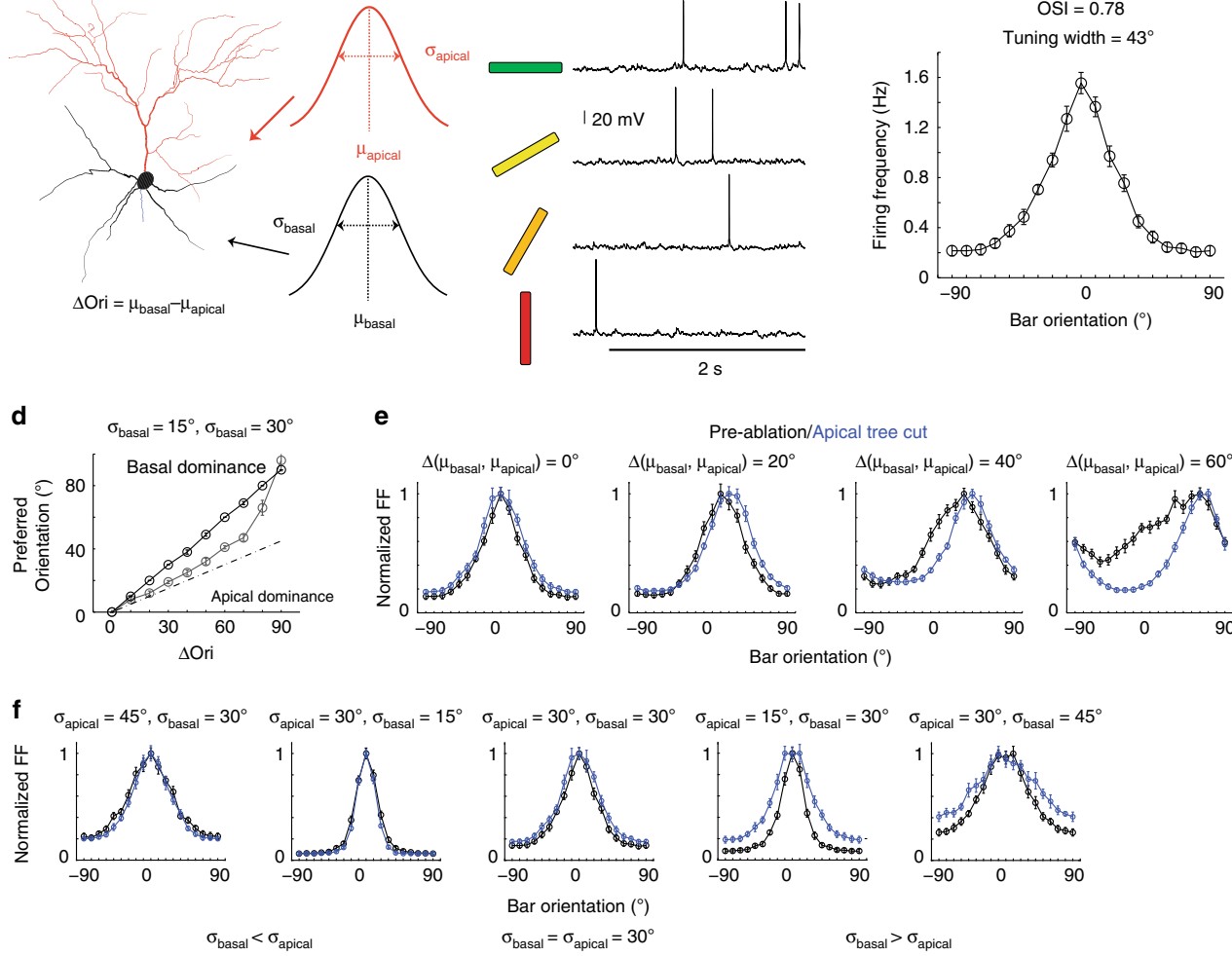

**Fig. 4** Biophysically realistic modeling of apical dendrite ablation. **a** Reconstruction of the L2/3 pyramidal neuron used for simulation. Three main parameters, standard deviation ($\sigma_{apical}$) of the distribution of $pref_{syn}$ in apical dendrites, standard deviation ($\sigma_{basal}$) and mean ($\mu_{basal}$) of the distribution of $pref_{syn}$ in basal dendrites, were varied. $\mu_{apical}$ is arbitrarily fixed at zero. **b** The model neuron generates realistic subthreshold and action potential activity in response to 0°, 30°, 60°, and 90° (left) orientations. Horizontal line represents stimulus duration (2 s). **c** The model neuron displays realistic tuning curves (average tuning curve of 10 simulations when $\sigma_{apical} = \sigma_{basal} = 30°$, $\mu_{apical} = \mu_{basal} = 0°$). **d** The model neuron's preferred orientation is biased to the basal dendrites' mean orientation preference for all $\Delta(\mu_{apical}, \mu_{basal})$. Black line, $\sigma_{basal} = 15°$. Gray line, $\sigma_{basal} = 30°$. For both conditions, $\sigma_{apical} = 30°$. Dotted line indicates equal contribution of apical and basal trees in determining somatic orientation preference. **e** Model neuron tuning curves pre- (black curve) and post- (blue curve) ablation of apical tree. Left two panels: At low $\Delta(\mu_{basal}, \mu_{apical})$ (<30°), ablation of the apical dendrite does not affect orientation tuning curve shape; Right two panels: At higher $\Delta(\mu_{basal}, \mu_{apical})$ (>30°), apical dendrite ablation leads to a shift in orientation preference and narrowing of tuning width. $\sigma_{apical} = \sigma_{basal} = 30°$. (**f**) Left two panels and the center: At $\sigma_{apical} > \sigma_{basal}$ or $\sigma_{apical} = \sigma_{basal} = 30°$, ablation of the apical dendrite does not affect the tuning width; Right two panels: At $\sigma_{apical} < \sigma_{basal}$, apical dendrite ablation broadens the tuning curve. $\Delta(\mu_{basal}, \mu_{apical}) = 0°$. Error bars in (**c**), (**e**), and (**f**) are standard error of the mean across simulations.

**Basal dendrite ablation: the shift vs. the drift hypothesis**. There are at least two possible explanations for the basal dendrite ablation results in Fig. 3. According to the shift hypothesis, the change in orientation preference following two-basal-dendrite ablation is due to a shift toward apical tree inputs assuming $\Delta(\mu_{basal}, \mu_{apical}) > 0°$. Computational analysis, removing basal dendrites from model neurons with different $\Delta$ and $\sigma$ (Supplementary Fig. 11i, j), weighs against the shift hypothesis. Although it was possible to generate model neurons with shifted orientation preference following two-basal-dendrite ablation (i.e., for $\Delta(\mu_{basal}, \mu_{apical}) = 30$–$50°$, $\sigma_{basal} \leq 30°$, $\sigma_{apical} = 15°$, mode change = 10°, Supplementary Fig. 11j), in these neurons apical ablation also led to a significant shift in orientation preference (Supplementary Fig. 11h), contrary to experiment. There was no parameter set for which the two-basal dendrite ablation, but not apical dendrite ablation, caused a shift in orientation preference.

The drift hypothesis entails that different basal dendrites are tuned to different preferences: cutting a subset of them will thus cause a drift due to loss of tuned basal inputs. To test this hypothesis, we generated model neurons with differentially tuned basal dendrites. The mean of the distribution of $pref_{syn}$ in each primary basal dendrite ($\mu_b$) was set independently. $\mu_b$ was selected from a range of $\Delta\mu$, where $\Delta\mu$ represents the maximum deviation of one basal dendrite from the mean orientation preference of the soma (here arbitrarily set to 0°, Fig. 5a). $\Delta\mu = 0°$ represents the null hypothesis, i.e., where all basal dendrites sample from the same salt-and-pepper arrangement of input orientation preferences centered at 0°. In this case, any post-ablation changes are due to sampling noise of the finite number of synapses and, as expected, basal dendrite ablation rarely causes changes in orientation preference (Fig. 5b, top row, Fig. 5d far left). The

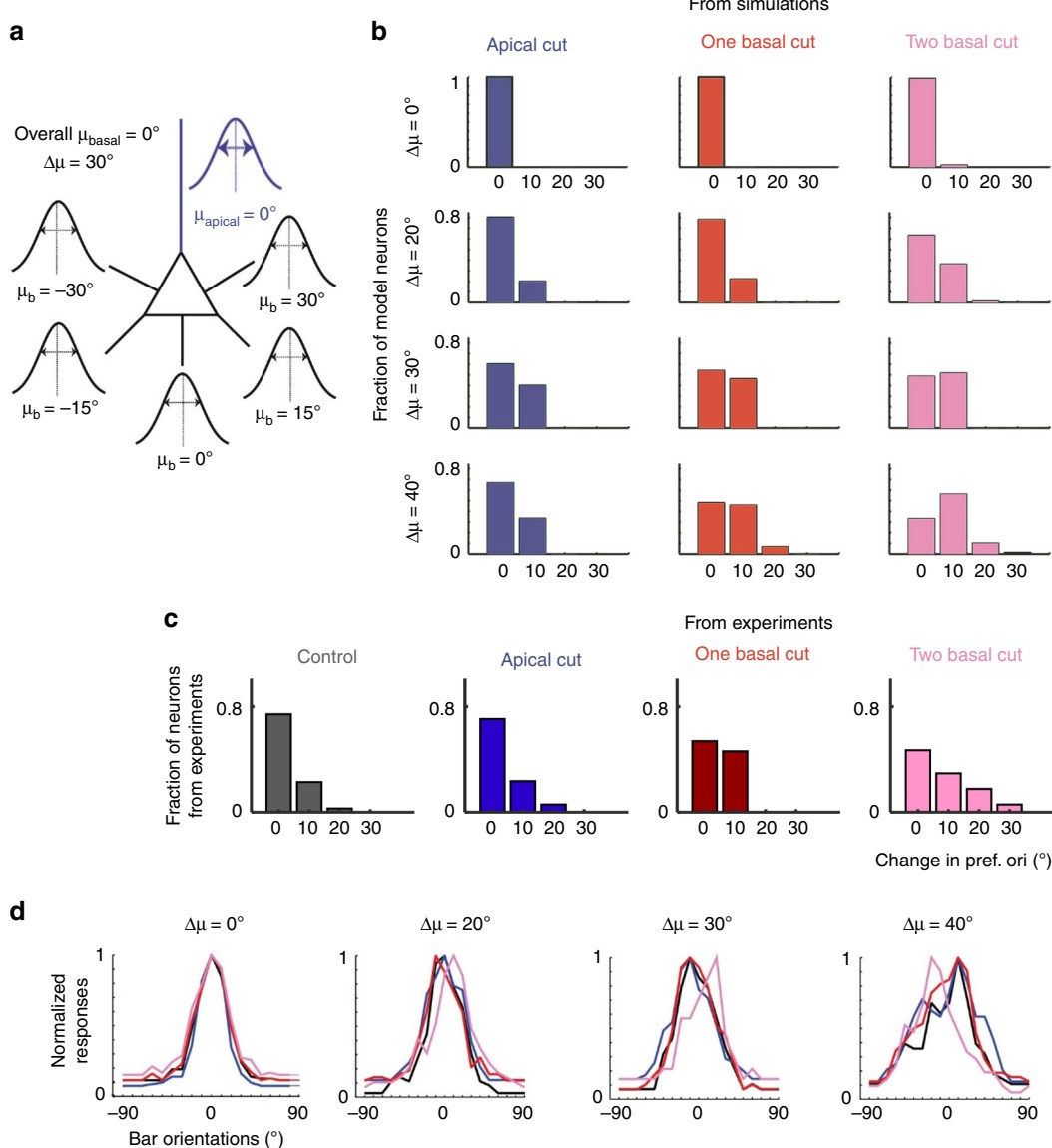

**Fig. 5** Simulated neurons with differentially tuned basal dendrites reproduce effects of ablation. **a** Example simulation schematic for disparity of orientation preference across basal dendrites, $\Delta\mu = 30°$. $\mu_b$ (mean orientation preference of tuned inputs of a given primary basal dendrite) was selected from a defined disparity $\mu_b \in \{-\Delta\mu \ldots \mu_{basal} \ldots \Delta\mu\}$. Mean of $pref_{syn}$ in the apical ($\mu_{apical}$) and basal ($\mu_{basal}$) were arbitrarily set to 0°. $\sigma_{apical} = 30°$ and $\sigma_b = \sigma_{basal} = 15°$ for all basal dendrites. **b** Increasing disparity between basal dendrites ($\Delta\mu$) leads to a mild shift in orientation preference with two-basal ablation while minimally affecting tuning curves of apical or one-basal ablation. Histogram of post-ablation absolute-value changes in orientation preference of model neurons with $\Delta\mu = 0°$ (same as Fig. 4), 20°, 30°, and 40° after apical (blue), single (red) and double (pink) basal dendrite ablation. For all, mean change in tuning-width is <10° and mean change in OSI is <0.2 (thresholds correspond to the mean + 1std of the experimental data). $\Delta\mu$ of 50° or greater fail to generate tuning curves comparable to experiment (OSI ≤ 0.2, tuning width ≥ 80° for more than 30% of simulated neurons). **c** In vivo experimental data in the format of (**b**), reproduced for comparison. **d** Example normalized tuning curves of model neurons with varying $\Delta\mu$ before (black) and after apical (blue), single (red) and double (pink) primary basal dendrite ablation.

same was true for small disparity values ($\Delta\mu = 20°$, Fig. 5b second row, Fig. 5d). Intermediate disparity values ($\Delta\mu = 30–40°$) appear to be necessary to achieve a modest change (mode = 10°, mean = 8°, max = 30° for $\Delta_\mu = 40°$) in orientation preference after two-basal-dendrite cut, while maintaining stability of orientation preference following apical or one-basal-dendrite cut, in agreement with experiment (Fig. 5b, c). Disparities ≥50° failed to generate tuning-curves compatible to the experimental data (OSI ≤ 0.2 and/or tuning-width ≥ 80°). As expected, the neuron's post-ablation changes in orientation preference strongly correlated with the magnitude of the change in mean orientation preference of the aggregates of the basal dendrites before and after

two basal dendrite ablation ($\widetilde{\Delta\mu}_{ablated}$, Supplementary Fig. 12). Figure 5 was generated using a narrow distribution of $pref_{syn}$ in basal dendrites ($\sigma_b = \sigma_{basal} = 15°$). $\sigma_b (= \sigma_{basal})$ of 30° (Supplementary Fig. 13) also induced a shift in orientation preference (max change = 20°, mean = 4.4°) but the fit to experimental data was worse than $\sigma_b = 15°$, indicating the need for sharply tuned input structures into basal dendrites.

In summary, these simulations suggest that to explain our experiment data, a) the difference in mean input tuning between apical and basal arbors, $\Delta(\mu_{basal}, \mu_{apical})$, must be small, b) the distribution of input tuning to the basal dendrites should be similar to or narrower than the distribution of synaptic input to

the apical tree ($\sigma_{basal} \leq \sigma_{apical} = 30°$) and c) there needs to be a degree of heterogeneity of tuning (30°–40°) across basal dendrites.

## Discussion

We showed that orientation selectivity in L2/3 pyramidal neurons, as measured by orientation preference, tuning-width and OSI, is robust to a complete loss of the apical tuft. This suggests that inputs to the basal dendritic domains are sufficient to determine the neuron's orientation selectivity[7,22]. This is surprising given that apical dendrites receive orientation-tuned input[11], with at least four input sources accessible to apical dendrites that could potentially modulate orientation selectivity in V1: (1) feedback cortico-cortical projections from extrastriate cortex thought to refine orientation selectivity[9,10,23,24], (2) direction-selective projections from the LGN[12], (3) thalamo-cortical projections from lateral posterior thalamus[13], and (4) dendrite-targeting inhibitory inputs[25,26]. Furthermore, several studies provided in vivo evidence of dendritic properties[27] in V1 apical dendrites that have the capacity to contribute to the neuron's orientation selectivity[6,28], one arguing that apical tuft input functions to narrow the neuron's tuning curve increasing selectivity[6]. However, these studies do not provide a causal connection between dendritic computations and the emergence of orientation selectivity at the soma. Our results suggest that with respect to orientation tuning apical dendritic tuft inputs are dominated by basal dendritic activity, at least under our experimental conditions, favoring a feedforward model for orientation selectivity[29,30]. Computational modeling confirmed that basal dendritic input is dominant over a wide range of synaptic input parameters. It is interesting to explore in the future what behavioral conditions[31–33] might change the relative contribution of apical dendritic inputs to orientation encoding in area V1.

Second, we showed that V1 neurons also remained robust to removal of 2 out of 5–8 primary basal dendrites. Specifically, only a small, though significant, change in orientation preference occurred in about 40% of the neurons that underwent two basal-dendrite ablation. The fact that we did not observe a marked loss of selectivity in any of the ablated neurons (Fig. 3) argues strongly against the possibility that there is a dominant, "master", dendrite[17,34], i.e., the situation in which input from a single dendrite dominates a neuron's properties. Based on our results, we estimate the probability of a master dendrite dominated neuronal output to be $p \approx 0.00009$ for neurons with six basal dendrites ($p \approx (5/6)^{13} \times (4/6)^{17} \approx 0.00009$ corresponding to 13 neurons with single and 17 neurons with double basal dendrite ablation), or $p \approx 0.0013$ for neurons with eight basal dendrites ($p \approx (7/8)^{13} \times (6/8)^{17} \approx 0.0013$).

Still, to explain the small but significant post-ablation change in orientation preference following two-basal dendrite ablation, simulations suggest that orientation tuning is somewhat heterogeneous across basal dendrites, with basal dendrite orientation preference spanning a range of ~30°–40° (Fig. 5). Such heterogeneity may arise from dendrite-specific forms of plasticity potentially mediated by spatially restricted biochemical signaling[35–37] and/or dendritic spikes[6,27,28,38–40].

We note that microdissection comes with some potential limitations: Damage to nearby neuronal processes due to ablation, though minimal, cannot be completely avoided; Astrocytes attracted to the ablation site post-ablation[15] could impact glutamate levels and modify the activity of the target neuron; Ion influx during the instant opening of the membrane in response to ablation could potentially induce unwanted plasticity on the connections that remain. Nonetheless our immunohistochemical findings showed that the damage around the ablation site is minimal (Supplementary Fig. 3). Furthermore our results show that neurons are active and remain tuned post-ablation, arguing that these processes are less likely to play a significant role.

These results give new insights into the complex structure-function relationship of the pyramidal neuron, the fundamental computational subunit of the neocortex. In particular, the remarkable robustness of orientation preference under dendritic micro-dissection hints at the extraordinary ability of sensory cortical neurons to maintain functional selectivity following input loss. Our approach emphasizes the importance of applying causal manipulations to study the contribution of dendritic arbors to sensory encoding. Dendritic microdissection is a powerful method for probing causal relations between dendritic structure/function and somatic properties that can be applied to several key questions in systems neuroscience research.

## Methods and data

**Animals in experiments**. All experimental protocols were approved by The Baylor College of Medicine (BCM) Institutional Review Board and Brigham and Women's Hospital (BWH) Institution Animal Care and Use Committee. Male and female wild-type (C57BL/6) mice were purchased from the institutional vivarium in BCM and Jackson Labs and bred for experiments.

**Chronic window implantation and sparse labeling**. Viral vectors were purchased from or edited/packaged by University of Pennsylvania Vector Core, Addgene and BCM Vector Core. For viral injections and chronic window implantation, 6–10 week old wild-type C57BL/6 mice (both male and female) were anaesthetized with isoflurane (1–1.5%). Baytril (5 mg kg⁻¹), Carprofen (5 mg kg⁻¹) and Dexamethasone (1.5 mg kg⁻¹) were administered subcutaneously. The depth of anesthesia was assessed via monitoring breathing rates. A headpost was implanted on the skull and a 3 mm diameter craniotomy was made over the visual cortex of the left hemisphere. The craniotomy was centered 2.7 mm lateral to the midline and 1.5 mm posterior to the bregma. To sparsely label pyramidal neurons with GCaMP6s, a mix (~90 nl per site, up to three sites) of diluted CamKII-CRE (AAV5 or AAV1, diluted 40,000–120,000×) and flex-GCaMP6s (AAV5, diluted up to 2×) (U Penn Vector Core, Addgene) or flex-mRuby-GCaMP6s (AAV8, BCM Vector Core) was injected slowly over 5 min per penetration using a Drummond Nanoject. Two to three penetrations ~0.5 mm apart on average were performed per craniotomy. This approach enabled (i) expression of GCaMP6s at sufficient levels in each cell to image the dendritic tree, while (ii) labeled neurons remained sparse. After the viral injection, a round coverslip was fitted to the craniotomy and sealed with vetbond and dental cement. Most chronic windows in our hands remain clear for 2–3 months. Visual stimulation and 2-photon imaging were performed on week 3–4 following viral injection, at which time GCaMP6s expression was optimal.

**In vivo calcium imaging of sparsely labeled neurons**. Three to four weeks after the injection, the GCaMP6s expressing mouse was sedated with Fentanyl (0.5 mg kg⁻¹) and Dexmedetomidine (0.5 mg kg⁻¹)[41] for imaging experiments. A stable level of anesthesia was confirmed by stable breathing rate and lack of movement. The right eye of the mouse was aligned to the center of the monitor. Customized light shielding was attached to the headpost to stimulate the mouse's right eye effectively without producing light artifacts in the two-photon images. We imaged calcium activity of ablation candidate neurons and their neighbor control neurons in L2/3 of mouse primary visual cortex during visual presentation. Images were acquired at ~9 frames/s using an Ultima IV microscope in spiral scanning mode with a 20×, 0.95 NA, Olympus objective or 25 × 1.0 NA Nikon objective (5–30 mW laser power, 900 nm). GCaMP6s gives excellent signal to noise ratio, corresponds well to the underlying firing rate, and

is well suited for measuring tuning functions. Orientation-tuning measurements and structure imaging were repeated before (day 0) and 1, 3, 5 days after ablation.

**In vivo dendrite ablation**. Under two-photon scanning, fluorescent dendrites were clear and visible at low laser power (<20 mW, 910 nm). Dendritic arbors of several L2/3 neurons were imaged using a custom Ultima IV 2-photon microscope. First, we screened for clearly orientation-tuned neurons with a clearly defined primary apical dendrite (Primary apical bifurcation > 20 μm away from the soma, soma depth between 150 and 250 μm, average pre-ablation orientation selectivity index for all ablated neurons was $0.79 \pm 0.18$, max = 0.99, min = 0.38). Note that these criteria likely excludes the L2 neurons commonly studied, e.g., in refs. [3,4,7]. Target apical dendrites were severed at least 15 μm away from the soma (Fig. 1c and Supplementray Movie 1, 2). Basal dendrites were severed at least 10 μm away from the soma (Fig. 3a, c and Supplementary Movie 4).

The ablation point was magnified 13–15 times with 1024 resolution at 910 nm, then ablated via repeated 200–400 ms point scans at 150–200 mW power, 800 nm wavelength. Approximate area impacted by single point scan is an ellipsoid of 0.4 μm (x,y) and 1.2 μm (z) diameter according to our point spread function (Supplementary Fig. 1a). Differences in ablation parameters depended on depth of ablation plane, overlying shadow casting vessels, and window clarity. As reported in previous studies (14–17), when a single point on a fluorescing dendrite received several focused laser pulses, the targeted dendrite formed a beads-on-a-string morphology immediately after the ablation and then degraded within a day (Supplementary Fig. 1). Successfully ablated dendritic segments become transiently brightly fluorescent as calcium enters the membrane, then recover its original brightness. Distal to the ablation point, dendritic segments displayed marked beading within ~1 h and degraded completely within a day. Dendritic segments proximal to the ablation point by about 5–10 μm survive indefinitely. Neurons that do not regain baseline fluorescence (~30% of ablated cells) disappear within 24 h. The remaining 70% of ablated neurons recover to baseline fluorescence levels within 2–3 h post-ablation. Of these, ~50% survive. This gives a net 30–40% survival rate for ablated neurons. For functionally mapped neurons, we found similar chance of survival rate both from apical (18 out of 39, from 26 mice) and basal dendrite ablation (13 out of 33 for one basal dendrite ablation and 17 out of 50 two basal dendrite ablation, from 34 mice), $\chi^2 = 1.1$, $p = 0.29$, chi-square test. Surviving neurons did not demonstrate significant regrowth or degradation at the ablation point (at most 3–5 μm regression toward the soma). The remaining dendritic arbor, as well as neighboring neuronal structures, maintained their original structure for at least 14 days following ablation (Fig. 1b, Supplementary Fig. 1b).

We assessed the possibility of injury to the neuron and/or neuropil in four ways. Evidence for minimal post-ablation damage includes: (1) ablated neurons that survived for a day lasted indefinitely after being stressed for a while (as documented by the calcium influx and increase in fluorescence, Supplementary Fig. 1d), (2) no visible change in spontaneous or visual-evoked calcium activity were observed in the days following ablation of either apical or basal dendrites (Figs. 1–3, Supplementary Figs. 4, 5), (3) no morphological changes were observed in non-ablated structures from the same neuron or processes of nearby non-ablated neurons, even 5 microns from the ablation site (Supplementary Fig. 1), and (4) immuno-labeled neuronal processes (anti-Tuj1) were minimally affected at the ablation site (Supplementary Fig. 3 and Supplementary Movie 3). Other labs using the same method have shown no effect of this ablation on

spine morphology or turnover in remnant proximal segments of ablated dendrites (36), and FIB-SEM reconstructions of ablation sites found a ~5 micron lesion with no prominent glial scar 5 days post-ablation[12]. Taking all of the data together leads us to conclude, as others have (refs. [14,15]), that the off-target damage mediated by 2-photon microdissection is minimal.

**Visual stimulation**. Visual stimuli were generated with MATLAB (Mathworks Inc.) PsychToolBox and presented on a Dell monitor (77° × 55° of visual angle), at a fixed mean luminance (80 candela/m2), positioned 32 cm in front of the animal. Prior to orientation mapping the animal was adapted for at least 15 min to the mean luminance level. To measure orientation selectivity, grayscale square-wave gratings (0.04 cycles/degree, 2 cycles/second) moving in one of 12 (30° steps) or 36 (10° steps) directions were presented in pseudorandom order over the full stimulation field. Stimulus presentation lasted 2 s and the inter-stimulus interval (uniform illumination set at the mean intensity) lasted 3 s. Twenty to thirty repetitions per stimulus direction of motion were collected and analyzed.

**Data analysis**. Calcium trace from each soma was selected and separated using custom MATLAB functions, including the ROI selecting function 'roigui' from T.W. Chen at Janelia Institute. Baseline fluorescence (F0) for calculating (F-F0)/F0 was the average of the 20% lowest values in a 20 sec (10 s pre, 10 s post) window around each frame. Visual-evoked calcium responses arose ~200 ms after the onset of the stimulus and peaked around the offset of the stimulus. If the s.e.m. (standard deviation of the mean) of peak responses (for the preferred orientation) overlapped with the s.e.m. of null orientation responses on Day 0 (before ablation), we excluded those neurons from analysis. We also excluded from ablation a few neurons that had multiple orientation preferences (their orientation tuning curve showed 2–3 peaks). Naturally, we also did not use such neurons as controls. The per-trial response was calculated as the average of a 1.5 s window centered at the peak of the mean response across all orientations. Tuning curves were calculated by averaging responses across repetitions of each stimulus orientation. From this tuning curve, orientation selectivity index was calculated with $(R_{pref} - R_{ortho})/(R_{pref} + R_{ortho})$. All orientation-tuning curves were fitted with von Mises function using Circstat tool box[42] after baseline subtraction. Preferred orientation and tuning width was calculated from the fitted tuning curve. Calculating preferred orientation and tuning width from the raw mean tuning curve generated similar results. To assess the effect of dendrite ablation on tuning curves, preferred orientation, tuning width and OSI values before ablation were subtracted from the corresponding values obtained 5 days after ablation. We used both $t$-test and Mann–Whitney $U$ test for pairwise comparisons and ANOVA and Kruskal–Wallis test for multi-group comparisons to assess how tuning values shift in ablated versus control neurons. Where necessary, Tukey test for multiple comparisons was used to compare between groups (e.g., control vs two-basal dendrite ablation) following ANOVA or Kruskal–Wallis test. Absolute-value was used to assess the shift in preferred orientation.

**Bootstrap analysis of orientation tuning reliability**. For the bootstrapping analysis in Supplementary Figs. 4–5, five single-trial responses were randomly sub-selected from the 20–50 trials acquired per orientation to generate simulated tuning curves. The preferred orientation was calculated from the von Mises-fit orientation tuning curve based on the sub-sampled data. This was repeated 1000 times to generate confidence intervals for the estimate of the preferred orientation pre- and post-ablation per

neuron. Distributions of the orientation estimates were shifted to have the mean of the preferred orientation estimates before ablation at 0°. A neuron was said to show significant difference in preferred orientation (or in tuning width) pre- versus post-ablation, if 95% of the confidence intervals established by bootstrap did not overlap with each other. The frequency of detecting neurons with significant difference in preferred orientation pre- and post- ablation was assessed using the chi-square test (Supplementary Figs. 4, 5, Supplementary Table 1).

**Immunofluorescence labeling**. Mice were killed by isoflurane overdose followed by cervical dislocation and decapitation. Following decapitation, brains were quickly removed and fixed in ice cold 4% PFA (pH 7.2, 1 h) followed by 20% sucrose immersion for 24 h and then 30% sucrose immersion for 24 h. Fixed brains were frozen in OCT medium and sectioned into 50μm coronal sections. For immunostaining, free floating sections were washed in 0.1 M PB and blocked in PBTGS (0.1 M PB with 0.3% Triton X-100 detergent and 5% goat serum) for 1 h. Sections were incubated in primary antibody over night at room temperature. The following antibodies were used: chicken anti-GFP (Abcam) and mouse anti-beta III Tubulin (Santa Cruz). Primary antibodies were then labeled using AlexaFluor-conjugated secondary antibodies (Invitrogen). Sections were visualized with a Zeiss Imager, Z1 fluorescent microscope with an Apotome attachment.

**Biophysical model**. The morphologically detailed L2/3 V1 pyramidal neuron model (Fig. 4a) was implemented in the NEURON simulation environment[43]. Model neurons (available in ModelDB,(accession number 231185) 3) were constrained against experimental electrophysiological and anatomical data[6,44–49] (see Supplementary information for detailed description). Synapses impinging in the apical and basal dendrites where either spontaneously activated or stimulus driven. Stimulus driven synapses were assigned with preferred orientations and randomly distributed along the dendrites[4]. When assigning orientation preference to synapses, we varied the standard deviation of the distributions ($\sigma_{basal}$, $\sigma_{apical}$), the difference ($\Delta$) of $\mu_{apical}$ and $\mu_{basal}$, as well as the $\mu_b$ of each individual basal dendrite ($\Delta\mu$). Control condition was the one corresponding to $\sigma_{apical} = \sigma_{basal} = 30°$, $\mu_{apical} = \mu_{basal} = 0°$, $\Delta\mu = 0°$, under which the OSI was 0.78 and the tuning width 43° (Fig. 4b, c). Ablation was simulated by removing the respective dendritic compartments and adjusting excitatory synaptic weights for the change in neuronal excitability. For each condition, we calculated the preferred orientation, tuning width and OSI of the resulting somatic tuning curves.

**Reporting summary**. Further information on research design is available in the Nature Research Reporting Summary linked to this article.

## Data availability
Raw datasets associated with Figs. 2 and 3 will be available from the corresponding authors on request. The biophysical model used during the current study is available in the ModelDB repository with the accession number 231185.

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

## Acknowledgements
S.S. and J.P. were supported by NEI R01 grant EY-024019 and NINDS R21 grant NS088457. A.P. and P.P. were supported by the European Research Council Starting Grant dEMORY (GA 311435). We thank Xiaolong Jiang for providing high quality Neurolucida reconstructions of L2/3 pyramidal neurons of mouse V1 visual cortex for the biophysical model study. We thank Matt Rasband for providing equipment and reagents for immunostaining experiments. We thank Sangkyun Lee for MATLAB expertise.

## Author contributions
J.P. and S.M.S. designed the study, A.P. and P.P. designed the modeling part of the study, J.P. and M.A.M. performed the experiments, J.P. analyzed the data, A.P. performed the simulations, J.P. and R.T.A. wrote the manuscript. J.P., R.T.A., A.P., P.P. and S.M.S. edited the manuscript together.

## Competing interests
The authors declare no competing interests.
