## [Peer Review File · Nature Communications]

Editorial Note: This manuscript has been previously reviewed at another journal that is not operating a transparent peer review scheme. This document only contains reviewer comments and rebuttal letters for versions considered at Nature Communications . Mentions of the other journal have been redacted.

Reviewers' Comments:

Reviewer #1:

Remarks to the Author:

All our points have been fully addressed, except for one. I am copying in our original point followed by the authors' reply.

2) Statistical treatment is unclear and potentially incorrect at times.

...

(F) On page 7, top paragraph (also see Supp Fig 6 (Supp Figure 9 in updated manuscript)) it says "we ablated the apical dendrite following 2-basal-dendrite ablation. This was successfully performed in two neurons, for which apical ablation did not further modify orientation preference", but you clearly cannot carry out statistics on $n = 2$. This means this conclusion cannot be drawn based on the data provided: "Thus, the orientation shift caused by 2-basal-dendrite cuts is unlikely to be due to a shift in the balance towards the apical tree's orientation preference." This data set thus has to be increased, or the section removed entirely.

...

> It is a matter of opinion whether it is appropriate or not to
> include this data. We agree with the reviewer that it would have
> been preferable to have a larger n number here, but the experiment
> is very difficult as there is a high chance of the neuron not
> surviving (see response to Minor Point 3 for detail), so we felt
> that it was not an efficient use of time to continue to attempt
> more triple-dendritic-arbor ablations. On the other hand, the
> example of even two neurons that maintain their tuning curve
> following triple-dendritic arbor ablation (i.e. loss of ~60% of the
> total input), strongly complements the statistically significant
> data about single- and double- dendritic ablation that we present
> in the paper and conveys significant information to the reader.
> These examples coupled to the single- and double- dendritic arbor
> ablation data do argue strongly about the robustness of the neuron
> to input loss, so we feel that the data should be included in the
> final manuscript to allow the readers to form their own opinions.
> We hope the reviewer will agree, but if not, we are happy to defer
> to the editors for deciding if they should be excluded.

No, it is not a matter of opinion. It is a matter of absence of statistics that is required to support the claim that is made. This requirement is what makes science science instead of opinion or guesswork.

We cannot recommend that this component of the paper be published in its present form. That the experiments are hard to carry out is not a good argument for pretending that the data set is complete, when in actuality it is not. An $n=3$ is a bare minimum for drawing a conclusion here, because that is the n that is required to carry out a t test to enable an actual comparison. But we would recommend that the n is increased even further. Or alternatively, if publication is urgent, this component of the paper could be removed.

Additionally, Supp Fig 8 is missing a caption title.

Reviewer #2:

Remarks to the Author:

The manuscript titled "Contribution of apical and basal dendrites of L2/3 pyramidal neurons to orientation encoding in mouse V1" by Park et al., was originally submitted to [Redacted], revised and resubmitted to Nature Communication. The subject of the study continues to be suitable for the interests of a broad scientific community. In fact, it is for this reason, I believe, that all three of the reviewers asked for some additional experiments or to increase the sample size to better understand the potential circuit mechanisms that underlie them in the previous round of reviews. I must say that I for one read the current manuscript with a slight disappointment that none of the suggested experiments were carried out and the request for increasing the sample size was not met. The authors are instead offering additional analyses to further support their initial findings, however, they themselves raise some concerns. As is, I cannot whole-heartedly support the publication of this manuscript in the journal. My comments on the manuscript as well as the rebuttal are as follows:

Major comments:

1) I am still concerned about the fact that the major conclusion of the study (that the somatic orientation tuning is robust to the loss of dendritic inputs) is drawn based on the assessment of the slightly over one-third of cases where the cell survived (42% for apical and 36% for basal ablation studies). It is easy to draw skepticism from the readers that there may be some link between the extent of the functional contribution of the ablated dendrite and the survival outcome of the neuron. If the reactive influx of calcium that resulted in the transient increase in the fluorescence upon dendritic ablation subsided within 5-90mins, have the authors tried to examine the post-ablation orientation tuning shortly after the 90min mark? If the neurons are recovered enough to be visually evoked, then direct comparison between pre and post-ablation as well as whether or not they survived subsequently would add some valuable information. Have you tried? Or are the neurons too sick?

2) It is unfortunate that the authors were unable to increase the sample size of their experiments. As also mentioned by Reviewer 1 in the previous round, the main conclusion that the ablation of two basal dendrites has the significant effect on the somatic orientation basically rides on a stat that is relatively weak in its significance ($p = 0.045$; control vs 2-basal), and the authors decision to address this concern by preprocessing the data by fitting subsampled data to curve before employing bootstrapping strategy seem to result in the overprocessing of the data (compare Sup Fig 4 and Sup Fig7). Selection of a data processing method based on the subsequent statistical significance is inappropriate.

3) Because of the lack of insights into the visually evoked dendritic activity and their response properties relative to somatic tuning, the conclusions drawn from the findings from the microdissection remains observational and the modeling speculative. In the rebuttal, the authors state that the basal dendrites are too deep to image, but what about the apical dendrites?

4) The authors seem to be highly critical of the other leading methods such as dendritic calcium imaging and direct dendritic patch recordings. For a balanced view, potential caveats of the rather invasive microdissection should be discussed.

Minor concerns:

1) page 2 "...while dendritic patch clamping,typically disrupts normal dendro-somatic processing": This is a strong statement against some of the leading studies that utilized this method. What is the basis for this assumption? It is rash to dismiss the entire field of dendritic patch clamp studies without evidence.

2) page 2 "The existence of functional inputs and activity on dendrites does not explain the necessity

of these inputs...": What is your alternative hypothesis then? That the neuron's final output is solely dependent on perisomatic inputs??

3) page 2 "Here we employed in vivo two photon microdissection to...": The way this introduction section is spun, I am not convinced that 2p microdissection is in anyway superior to dendritic patch clamping or dendritic calcium imaging (input mapping). In fact, it is the most invasive method of the three.

4) page 4 "...but it is not clear whether these contribute to the neuron's..." : Change to "...but is it not clear how..."

5) page 5 last sentence "This demonstrates that... inputs to the basal dendrites are sufficient...": This conclusion seems out of place as the effects of basal dendrite ablations have not yet been discussed at this point in the text".

6) page7 "To test the possibility that the shift in orientation preference following 2-basal-dendrite ablation might be...": As pointed out by one of the reviewers, $n = 2$ is not statistically sufficient to "test a possibility". Thus, this entire paragraph does not extend beyond a speculative interpretation. The authors must either strip this down to plainly stating their observation and save their interpretation for the discussion section or remove this paragraph entirely. Similarly on page 9, "Our exploratory observation that..." is still attempting to draw conclusion from the observation based on $n = 2$. Please omit or move to discussion.

7) page11 "neurons can still compute orientation tuning without those regenerative dendritic electrical events...": Ref 30 cited leading up to this sentence was a study done in layer 5 and Purkinje cells, two cell types both morphologically and functionally distinct from the layer 2/3 neuron studied here. The authors do not have any data to specifically address the effects of the dendritic spikes on somatic tuning. Please remove.

Regarding the rebuttal to my (Rev2) concerns:

1) Selection by survival: my lingering concern regarding this issue is expressed above under major concern 1).

2) Using layer 2/3 neurons to study apical vs basal dendritic contributions: the response by the authors clarified to me that their criterion specifically excludes the L2 neurons commonly studied in the refs 3, 4, 8, and 19. Perhaps it is helpful to explicitly state this in the text.

3) Sample size issue: Please see below under Response to other reviewers point 2).

4) Regarding the trend of the direction of change in the response amplitude, the authors point to the "unnormalized" polar plot in Figure 2C and say that there was a trend (not statistically significant) for the response amplitude to decrease after the apical dendrite ablation. Then what am I looking at in the raw traces on the right side of Figure 2a? If these traces are normalized, then the scale must indicate as such.

5) OK.

Adequacy of the response to other reviewers (as requested by the editor)

1) Regarding the concern of drawing conclusions based on $n = 2$ (Reviewer 1): I agree with the authors assertion that it is a matter of opinion whether it is appropriate or not to include the given

data set. It is not, however, a matter of opinion, whether a scientific conclusion should be drawn from observation based on limited sample size that are not statistically sufficient: It should not. Therefore, I agree with the reviewer's assessment. Please see my comments above regarding this issue.

2) Issue of sample size (raised by Rev 1): I don't think changing the method for data preprocessing is an adequate way to address the concern and misses the original point that the sample size is small.

Reviewer #3:

Remarks to the Author:

As I mentioned in my first review for [Redacted], this study is an original and technically creative approach to determine the impact of apical and basal dendrites on the orientation selectivity of layer 2/3 pyramidal cells. The authors clearly demonstrate that ablation of the apical dendrite or up to two basal dendrites has no or only little effect on the orientation selectivity of these neurons. The new version of manuscript is in my opinion improved and good for publication. I have no more comments.

Reviewers' comments:

Reviewer #1 (Remarks to the Author):

All our points have been fully addressed, except for one. I am copying in our original point followed by the authors' reply.

2) Statistical treatment is unclear and potentially incorrect at times.

(F) On page 7, top paragraph (also see Supp Fig 6 (Supp Figure 9 in updated manuscript)) it says "we ablated the apical dendrite following 2-basal-dendrite ablation. This was successfully performed in two neurons, for which apical ablation did not further modify orientation preference", but you clearly cannot carry out statistics on $n = 2$. This means this conclusion cannot be drawn based on the data provided: "Thus, the orientation shift caused by 2-basal-dendrite cuts is unlikely to be due to a shift in the balance towards the apical tree's orientation preference." This data set thus has to be increased, or the section removed entirely.

> It is a matter of opinion whether it is appropriate or not to include this data. We agree with the reviewer that it would have been preferable to have a larger n number here, but the experiment is very difficult as there is a high chance of the neuron not surviving (see response to Minor Point 3 for detail), so we felt that it was not an efficient use of time to continue to attempt more triple-dendritic-arbor ablations. On the other hand, the example of even two neurons that maintain their tuning curve following triple-dendritic arbor ablation (i.e. loss of ~60% of the total input), strongly complements the statistically significant data about single- and double- dendritic ablation that we present in the paper and conveys significant information to the reader. These examples coupled to the single- and double-dendritic arbor ablation data do argue strongly about the robustness of the neuron to input loss, so we feel that the data should be included in the final manuscript to allow the readers to form their own opinions. We hope the reviewer will agree, but if not, we are happy to defer to the editors for deciding if they should be excluded.

No, it is not a matter of opinion. It is a matter of absence of statistics that is required to support the claim that is made. This requirement is what makes science science instead of opinion or guesswork.

We cannot recommend that this component of the paper be published in its present form. That the experiments are hard to carry out is not a good argument for pretending that the data set is complete, when in actuality it is not. An $n=3$ is a bare minimum for drawing a conclusion here, because that is the n that is required to carry out a t test to enable an actual comparison. But we would recommend that the n is increased even further. Or alternatively, if publication is urgent, this component of the paper could be removed.

We are happy to defer to the reviewers' opinion/reasoning. We therefore removed all references to the three-dendrite ablation pilot results from the results and discussion. We would like, if possible, to leave the actual experimental data obtained as a supplementary data figure, in the interest of fully disclosing all experiments that were performed for the paper. However, we are happy to defer to the final opinion

of the editor and withdraw these data even from the supplementary material, if that is deemed appropriate.

Additionally, Supp Fig 8 is missing a caption title.

We appreciate the reviewer's keen eye and corrected the Supp Fig 8 accordingly in the new manuscript.

Reviewer #2 (Remarks to the Author):

The manuscript titled "Contribution of apical and basal dendrites of L2/3 pyramidal neurons to orientation encoding in mouse V1" by Park et al., was originally submitted to [Redacted], revised and resubmitted to Nature Communication. The subject of the study continues to be suitable for the interests of a broad scientific community. In fact, it is for this reason, I believe, that all three of the reviewers asked for some additional experiments or to increase the sample size to better understand the potential circuit mechanisms that underlie them in the previous round of reviews. I must say that I for one read the current manuscript with a slight disappointment that none of the suggested experiments were carried out and the request for increasing the sample size was not met. The authors are instead offering additional analyses to further support their initial findings, however, they themselves raise some concerns. As is, I cannot whole-heartedly support the publication of this manuscript in the journal. My comments on the manuscript as well as the rebuttal are as follows:

We now performed additional experiments, as the reviewer #2 suggested, and added an additional 7 neurons with apical dendrite ablation and 3 neurons with two basal dendrite ablation which further improves the overall statistical power. Please note that the bootstrap analysis the reviewer refers to above was added in our previous response to comply with a prior suggestion by reviewer 1. We respond in more detail below:

Major comments:

1) I am still concerned about the fact that the major conclusion of the study (that the somatic orientation tuning is robust to the loss of dendritic inputs) is drawn based on the assessment of the slightly over one-third of cases where the cell survived (42% for apical and 36% for basal ablation studies). It is easy to draw skepticism from the readers that there may be some link between the extent of the functional contribution of the ablated dendrite and the survival outcome of the neuron. If the reactive influx of calcium that resulted in the transient increase in the fluorescence upon dendritic ablation subsided within 5-90mins, have the authors tried to examine the post-ablation orientation tuning shortly after the 90min mark? If the neurons are recovered enough to be visually evoked, then direct comparison between pre and post-ablation as well as whether or not they survived subsequently would add some valuable information. Have you tried? Or are the neurons too sick?

Unfortunately, the baseline fluorescence of target neurons 5-90min post-ablation is still high and the cells' dF/F are poorly modulated by visual stimuli (i.e. dF/F signal is noisy). This suggests GCaMP is still partially saturated and the neuron is not fully recovered at this time point. In early ablation experiments we performed under whole cell patch, we observed similar noisy visual responses within 1-1.5 hours

post ablation, and this led us to abandon this method in favor of chronic calcium imaging. To further address the reviewer's question, we performed additional experiments monitoring four neurons 90 min to 24 hours post ablation. Cell fluorescence levels started to recover (i.e. decrease) within 1.5-6 hours post ablation, normal appearing spontaneous activity was visible by 15 hours post ablation, and by 24 hours all surviving neurons responded to visual stimuli and showed the orientation tuning similar to what was observed before ablation. It is unlikely that persistent neuronal injury plays a role in the results we present from day 5 post ablation (analysis done on day 3, or day 1 post ablation yields similar results).

We have no reason to believe that the likelihood of cell death post-ablation depended on the functional properties of the neuron or the targeted dendrite. The likelihood of cell loss post-ablation did not depend on the morphology or extent of the ablated dendrite, nor did it depend on the cell's baseline fluorescence. The major risk factors for post-ablation cell loss/death that we could identify are the ablation power used and number of point scans required to disconnect the dendrite. More point scans at higher power were required to ablate dendritic segments when the target dendritic point was located closer to a large occluding blood vessel and/or the optical clarity of the window was compromised, and these cells received more point scans at higher power were more likely to disappear in the days post-ablation.

2) It is unfortunate that the authors were unable to increase the sample size of their experiments. As also mentioned by Reviewer 1 in the previous round, the main conclusion that the ablation of two basal dendrites has the significant effect on the somatic orientation basically rides on a stat that is relatively weak in its significance ($p = 0.045$; control vs 2-basal), and the authors decision to address this concern by preprocessing the data by fitting subsampled data to curve before employing bootstrapping strategy seem to result in the overprocessing of the data (compare Sup Fig 4 and Sup Fig7). Selection of a data processing method based on the subsequent statistical significance is inappropriate.

To address this, we added an additional 7 neurons with apical dendrite ablation and 3 neurons with two basal dendrites ablation. The additional data improved the statistics for basal dendrite ablation: on average, the shift in orientation preference after two basal dendrite ablation is bigger than that of one basal dendrite ablation or control neurons ($p=0.003$, ANOVA with Tukey correction for multiple comparisons; Kruskal-Wallis output with Tukey correction for multiple comparisons: $p=0.025$ for the comparison control vs 2-basal). The small shift upon 2-basal dendrite ablation is more significant when 2-basal data is compared to pooled data of 1-basal and controls, groups of neurons that did not change the preferred orientation ($p=0.009$, Kruskal-Wallis). Seven additional neurons with apical dendrite ablation also strengthens our previous finding: there is no significant difference in orientation preference change between control and apical dendrite ablation ($p=0.4$, t-test; $p=0.95$, $\chi^2(1,33)=0$, Kruskal-Wallis). This addresses Reviewer 1's concerns about the significance of apical ablation results.

To clarify, our strategy using bootstrapped estimates of preferred orientations (Supp. Fig 4-5 and Supp. Table 1) to test if orientation preference is stable after dendrite ablation was in response to reviewer #1's previous suggestion: "I suggest increasing the n overall, describing the statistical treatment better, and also improving the statistical treatment, perhaps by combining parametric and non-parametric

tests, side by side, perhaps by using some sort of bootstrap Monte Carlo approach to simulate significance with the kinds of truncated normal distributions that the authors are likely getting by taking the absolute value of a difference in the delta-ORI.”

Supp. Fig. 4 illustrates the distributions of bootstrapped estimates of preferred orientations per neuron. It is not comparable to Supp Fig 7 which shows orientation tuning curves before and after ablation. For detailed method of Supp. Fig. 4, see associated figure legend and Data Analysis: Bootstrap simulation analysis of orientation tuning reliability section of Methods. The width of this distribution indicates how reliable the measurement of preferred orientation preference is. If the 95% confidence interval of the pre-ablation orientation preference distribution overlaps with the one obtained post-ablation, it suggests that a potential shift in orientation preference is more likely to result from random fluctuations. Supp Fig 4 shows that the shift of orientation preference upon two basal-dendrite ablation is more likely to be an effect of the ablation treatment versus from random fluctuations, ~40% (7 out of 17) of the neurons that underwent this treatment ($p=0.0015$, Chi-square test, Supp. Table 1).

3) Because of the lack of insights into the visually evoked dendritic activity and their response properties relative to somatic tuning, the conclusions drawn from the findings from the microdissection remains observational and the modeling speculative. In the rebuttal, the authors state that the basal dendrites are too deep to image, but what about the apical dendrites?

Dendritic activity relative to somatic tuning has been an exciting and active field of study so far. Numerous studies have demonstrated visually evoked dendritic activity: multiple orientation tuned inputs are scattered across the apical dendritic tree (Chen et al., 2013; Iacaruso, Gasler, & Hofer, 2017; Jia, Rochefort, Chen, & Konnerth, 2010) (Refs 3, 4 and 11 of our manuscript). Chen et al imaged calcium activities of dendritic spines and showed that inputs on apical dendritic trees are biased toward the soma's preferred orientation. These pioneering studies provided the starting point of our study, which was to test if the removal of those inputs on dendrites and corresponding dendritic functional apparatus would impact the neuron's final output from the soma. Rather than repeating already duplicated findings, we chose to adopt the published observations to inform the input structure of our model neurons, and to explore which possible input structure agrees with both our experiment data as well as with previously published observations. We agree with the reviewer that in the future it is important to design more detailed studies that ideally combine ablation with dendritic calcium response imaging.

4) The authors seem to be highly critical of the other leading methods such as dendritic calcium imaging and direct dendritic patch recordings. For a balanced view, potential caveats of the rather invasive microdissection should be discussed.

As the reviewer points out, microdissection comes with caveats: 1. Damage to nearby neuronal processes due to ablation, though minimal, cannot be completely avoided, 2. Astrocytes attracted to the ablation site post ablation (Canty et al., 2013) (Ref 15) could impact glutamate levels and modify the activity of the target neuron, 3. Ion influx during the instant opening of the membrane in response to ablation could potentially induce unwanted plasticity on the connections that remain. Nonetheless our immunohistochemical findings showed that the damage around the ablation site is minimal (Supp. Fig.

3). Furthermore our results show that neurons are active and remain tuned, arguing that these processes are less likely to play a significant role (since it is highly unlikely that they would contrive to cancel exactly the changes induced by ablation).

We now discuss the caveats of the microdissection in the revised discussion as well.

Minor concerns:

1) page 2 "...while dendritic patch clamping, ...typically disrupts normal dendro-somatic processing": This is a strong statement against some of the leading studies that utilized this method. What is the basis for this assumption? It is rash to dismiss the entire field of dendritic patch clamp studies without evidence.

It was not our intention to dismiss this important and scientifically productive field. We changed the sentence to "while *in vivo* dendritic patch clamping, despite its power at dissecting specific hypotheses, is usually limited to assessing single branch contributions to neuronal responses".

2) page 2 "The existence of functional inputs and activity on dendrites does not explain the necessity of these inputs...": What is your alternative hypothesis then? That the neuron's final output is solely dependent on perisomatic inputs??

We apologize for the poor wording. It was not our intent to argue that. Rather, what we meant to say is that because there are orientation tuned inputs available on apical dendrites (Chen et al., 2013; Iacuruso et al., 2017; Jia et al., 2010), one may naturally hypothesize that the apical inputs should contribute strongly to the soma's final output, in the sense that if they are eliminated orientation preference would be expected to change. Our experimental data, in contrast, suggests that a neuron can maintain its orientation selectivity without the inputs on the apical dendrites.

3) page 2 "Here we employed *in vivo* two photon microdissection to...": The way this introduction section is spun, I am not convinced that 2p microdissection is in anyway superior to dendritic patch clamping or dendritic calcium imaging (input mapping). In fact, it is the most invasive method of the three.

There are many different valid and valuable ways of studying what inputs dendrites receive and how they influence the soma's final output. We agree with the reviewer that dendritic patch clamping or dendritic calcium imaging is a great experimental strategy to locate and map inputs available on dendrites. However, the existence of the inputs does not necessarily immediately determine how crucial they are for generating specific functional outputs at the soma. Ablation is a causal method that adds an interesting experimental dimension to this analysis. As we responded in Reviewer 2's major comment 4, We discuss limitations arising from the impact that ablation has on the target neuron and nearby processes in the results, methods, and revised discussion sections (see also Supp Fig 3 and the EM study in Ref 15 of our manuscript (Canty et al., 2013)). Interestingly, and even surprisingly, our results clearly show that neuronal activity and orientation tuning are minimally affected by microdissection of even large portions of the dendritic tree.

4) page 4 "...but it is not clear whether these contribute to the neuron's..." : Change to "...but is it not clear how..."

We changed the sentence accordingly.

5) page 5 last sentence "This demonstrates that.... inputs to the basal dendrites are sufficient...": This conclusion seems out of place as the effects of basal dendrite ablations have not yet been discussed at this point in the text".

What we wanted to convey to the audience was that the remained inputs after the removal of apical dendrites are sufficient for a pyramidal neurons to maintain orientation selectivity. We feel that this sentence as written serves as a good transition leading into the reporting of the basal dendrite ablation data that immediately follows. If the editor agrees that this correction is necessary, we will modify the sentence as suggested in the final version.

6) page7 "To test the possibility that the shift in orientation preference following 2-basal-dendrite ablation might be...": As pointed out by one of the reviewers, $n = 2$ is not statistically sufficient to "test a possibility". Thus, this entire paragraph does not extend beyond a speculative interpretation. The authors must either strip this down to plainly stating their observation and save their interpretation for the discussion section or remove this paragraph entirely. Similarly on page 9, "Our exploratory observation that..." is still attempting to draw conclusion from the observation based on $n = 2$. Please omit or move to discussion.

As the reviewer suggested, we removed all sentences discussing our pilot findings of orientation tuning stability following 3-basal-dendrite ablation ($n=2$) and apical + two-basal-dendrite ablation ($n=2$) from the manuscript. We would like, if possible, to leave the actual experimental data obtained as a supplementary data figure, in the interest of fully disclosing all experiments that were performed for the paper. However, we are happy to defer to the final opinion of the editor and withdraw these data even from the supplementary material, if that is deemed appropriate.

7) page11 "neurons can still compute orientation tuning without those regenerative dendritic electrical events...": Ref 30 cited leading up to this sentence was a study done in layer 5 and Purkinje cells, two cell types both morphologically and functionally distinct from the layer 2/3 neuron studied here. The authors do not have any data to specifically address the effects of the dendritic spikes on somatic tuning. Please remove.

We removed the sentence as suggested.

Regarding the rebuttal to my (Rev2) concerns (from Reviewer 2):

1) Selection by survival: my lingering concern regarding this issue is expressed above under major concern 1).

We shared our experiment results and thoughts in response to the Reviewer 2 major concern/comment 1 above, page 3-4 of this rebuttal letter.

2) Using layer 2/3 neurons to study apical vs basal dendritic contributions: the response by the authors clarified to me that their criterion specifically excludes the L2 neurons commonly studied in the refs 3, 4, 8, and 19. Perhaps it is helpful to explicitly state this in the text.

This clarification was added to the Method section, under 'in vivo dendrite ablation': "First, we screened for clearly orientation-tuned neurons with a clearly defined primary apical dendrite (Primary apical bifurcation >20 μm away from the soma, soma depth between 150 and 250 μm ...). Note that these criteria likely excludes the L2 neurons commonly studied, e.g. in refs 3, 4, and 20."

3) Sample size issue: Please see below under Response to other reviewers point 2).

We added 7 additional neurons with apical dendrite ablation and 3 additional neurons with two basal dendrite ablation to the manuscript, which addresses concern 3 of Reviewer #2.

4) Regarding the trend of the direction of change in the response amplitude, the authors point to the "unnormalized" polar plot in Figure 2C and say that there was a trend (not statistically significant) for the response amplitude to decrease after the apical dendrite ablation. Then what am I looking at in the raw traces on the right side of Figure 2a? If these traces are normalized, then the scale must indicate as such.

This question reflects a very close examination of the figure fidelity for which we are highly grateful. In fact, the responses that we show (to 90° gratings) are not notably changed between day 0 and day 5 in this example neuron. The more pronounced change in gain is seen for other orientations, particularly the opposite (270°) orientation. This can be seen on the un-normalized polar plots in Fig. 2c.

5) OK.

Adequacy of the response to other reviewers (as requested by the editor)

1) Regarding the concern of drawing conclusions based on $n = 2$ (Reviewer 1): I agree with the authors assertion that it is a matter of opinion whether it is appropriate or not to include the given data set. It is not, however, a matter of opinion, whether a scientific conclusion should be drawn from observation based on limited sample size that are not statistically sufficient: It should not. Therefore, I agree with the reviewer's assessment. Please see my comments above regarding this issue.

As the reviewer suggested, we removed all sentences discussing our pilot findings of orientation tuning stability following 3-basal-dendrite ablation ($n=2$) and apical + two-basal-dendrite ablation ($n=2$) from the manuscript. We would like, if possible, to leave the actual experimental data obtained as a supplementary data figure, in the interest of fully disclosing all experiments that were performed for the paper. However, we are happy to defer to the final opinion of the editor and withdraw these data even from the supplementary material, if that is deemed appropriate.

2) Issue of sample size (raised by Rev 1): I don't think changing the method for data preprocessing is an adequate way to address the concern and misses the original point that the sample size is small.

We added 7 additional apical dendrite ablation and 3 additional two-basal ablation neurons to the

existing data, which strengthens our original results. Please see our response to the Reviewer #2's major concern No.2

Reviewer #3 (Remarks to the Author):

As I mentioned in my first review for [Redacted], this study is an original and technically creative approach to determine the impact of apical and basal dendrites on the orientation selectivity of layer 2/3 pyramidal cells. The authors clearly demonstrate that ablation of the apical dendrite or up to two basal dendrites has no or only little effect on the orientation selectivity of these neurons.

The new version of manuscript is in my opinion improved and good for publication. I have no more comments.

Reviewers' Comments:

Reviewer #1:

Remarks to the Author:

The authors have addressed our remaining comment. This very nice study is now ready for publication in Nature Communications!

Reviewer #2:

Remarks to the Author:

All of my previous concerns have been addressed, and I have no further comments. As for the rebuttal concerning my previous points #5 and 6, I am happy to leave them up to the editorial decision as suggested by the authors.